# Ternary Epoxy Nanocomposites with Synergistic Effects: Preparation, Properties Evaluation and Structure Analysis

**DOI:** 10.3390/polym17020158

**Published:** 2025-01-10

**Authors:** Patryk Suroń, Anita Białkowska, Mohamed Bakar, Barbora Hanulikova, Milan Masař, Dora Kroisová

**Affiliations:** 1Faculty of Applied Chemistry, University of Radom, 26-600 Radom, Poland; a.bialkowska@uthrad.pl (A.B.); mbakar@wp.pl (M.B.); 2Centre of Polymer Systems, Tomas Bata University in Zlin, 760 01 Zlin, Czech Republic; hanulikova@utb.cz (B.H.); masar@utb.cz (M.M.); 3Faculty of Mechanical Engineering, Technical University of Liberec, 461 17 Liberec, Czech Republic; dora.kroisova@tul.cz

**Keywords:** epoxy hybrids, mechanical properties, synergism, thermal properties, structure and morphology

## Abstract

The objective of the present work was to prepare hybrid epoxy composites with improved mechanical and thermal properties. The simultaneous use of two different modifiers in an epoxy resin was motivated by the expected occurrence of synergistic effects on the performance properties of the matrix. Such a hybrid composite can be used in more severe conditions and/or in broader application areas. Hybrid epoxy composites were prepared with polyurethane (PUR), Nanomer nanoclay and carbon nanotubes (CNT), followed by the evaluation of their mechanical and thermal properties. Synergistic improvements in mechanical properties of hybrid composites were observed for 0.5 wt% Nanomer and 1 wt% carbon nanotubes (CNT), 7.5 wt% PUR and 1 wt% CNT, and 5 wt% PUR and 1 wt% CNT, confirming the occurrence of synergistic effects as to the impact strength (IS) of the matrices, compared to binary systems. The toughening induced by CNT/Nanomer modifiers can be attributed to the specific interfacial interactions between the two nanoparticles, while in the case of CNT/PUR, it can be explained by the combined effects of flexible polymer chains and the specific arrangement of nanoparticles in epoxy systems. Spectroscopy analysis confirmed the occurrence of interaction between OH groups in the epoxy matrix with CNT and reactive groups of PUR. The fracture surface showed plastic deformations, with good dispersion of CNT, explaining the improved mechanical properties of the matrix composites.

## 1. Introduction

Hybrid polymer nanocomposites have received, in recent decades, considerable attention from both academics and industrialists because of the improved mechanical performance, thermal stability and barrier properties of these materials [1,2,3,4,5,6,7,8,9,10]. The unique properties of hybrid composites are the result of the interaction of integral components of the composition according to various strengthening mechanisms. This often leads to the synergism of properties in these hybrid materials. The phenomenon of exfoliation and/or intercalation of solid nanoclay nanoplatelets, the formation of a second phase by soft nanoparticles or the formation of a thermoplastic with possible grafting reactions with the matrix would explain the improvement in the toughness of the latter.

These hybrid polymer composite materials can be prepared using various matrices and a large group of different modifiers such as solid microparticles, nanoparticles, liquid rubbers, plasticizers or thermoplastics. The choice of matrix in composites is dictated by its properties, the price and the ease of processing. Epoxy resin is one of the cheaper matrix materials, and it demonstrates good processing properties. However, its brittleness and low resistance to crack-propagation make its modification necessary. Modifiers have recently been widely used, either alone or combined with other modifiers, to prepare hybrid epoxy composites with improved mechanical and thermal properties. Recently, interesting reviews [11,12,13,14,15,16,17] and scientific works [18,19,20,21,22,23,24,25,26] have been conducted using two modifiers in order to obtain synergistic effects from the properties of epoxy hybrids, thus demonstrating the importance of the present investigation.

Carbon nanotubes (CNTs) are known for their excellent stiffness and strength, and thus they have been considered as potential modifiers for various polymers. They have been combined with, among others, different engineering thermoplastics, nanoclays and graphene to form hybrid composites with epoxy resin.

Polyetherimide (PEI) and polyether sulphone (PES) were combined with carbon nanotubes (CNTs) to prepare hybrid epoxy nanocomposites with synergistic toughening. Chen et al. [18] simultaneously used PEI and multi-walled carbon nanotubes modified with amine groups (NH_2_-MWCNTs) to improve the fracture toughness K_C_ of diglycidyl ether of bisphenol A (DGEBA). The toughening was explained by the crack-deflection and bridging properties of the plastic deformation of PEI, and the pulling forces of the carbon nanotubes. However, the use of PEI with carboxyl-functionalized multiwalled carbon nanotubes (COOH-MWCNTs) led to a synergistic effect on the fracture toughness of DGEBA which was attributed to the phase separation of PEI, good dispersion of CNTs in the matrix and active crack-energy dissipation [19]. A new method to reinforce epoxy resin was introduced by the use of MWCNT which was previously attached to exfoliated montmorillonite (MMT) [20]. Synergistic toughening of the epoxy resin was achieved at a MWCNT/MMT weight ratio of 0.1:1, due to strong interfacial adhesion between the MWCNT/MMT and the matrix, as well as the uniform stress distribution.

Carbon nanotubes were also combined with polyetheretherketone (PEEK) and thermoplastic polyetherketone-cardo (PEK-C) to produce hybrid epoxy (EP). The results obtained showed significant improvements in the strength and toughness of the matrix [21], while the stronger interaction between PEEK/CNT and EP resulted in lower friction coefficients for the coatings [22]. However, the results of EP/CNT/PEK-C hybrid [23] confirmed that the fracture toughness was closely related to the two-phase structure formed between EP and PEK-C.

Hyperbranched polymers (HBPs) and ultrahigh-molecular-weight polyethylene (UHMWPE) showed significant improvement in the properties of the epoxy matrix when combined with MWCNTs [24,25]. The impact strength, tensile strength, tensile modulus, fracture toughness, and glass transition temperature of the matrix were significantly increased compared to binary systems and pure resin [24]. However, the work of adhesion, tensile strength and tensile modulus of the epoxy matrix were enhanced by ~26%, ~67% and ~35%, respectively, with UHMWPE [25].

Other studies have focused on the preparation of ternary epoxy nanocomposites combining CNTs with polycarbonate [26] and CNTs with branched polyethyleneimine [27]. In both cases, the properties of the matrix were improved to the good dispersion of the modifiers and interfacial interactions.

The concomitant addition of graphene oxide (GO) and CNTs resulted in improvements in the tensile strength, critical stress intensity factor (K_C_), and critical strain energy release rate (G_C_) of the hybrid composite, due to the good dispersion of the nanofillers in the matrix [28]. However, Chatterjee et al. [29] confirmed that larger graphene nanoplatelets (GnPs)-based nanoparticles led to greater improvement in fracture toughness of the epoxy resin–modified CNTs (by ~75%, compared to the pristine matrix). Ghaleb et al. [30] confirmed that maximum improvements in the tensile and electrical properties of an epoxy matrix modified with a graphene nanopowder (GNP) and MWCNT hybrid nanocomposite was achieved with a GNP/MWCNT ratio of 0.1:0.4. The uniform dispersion of the nanoparticles in the matrix, combined with the alignment of the MWCNTs on the surface of the GNP, led to this improvement. A similar level of improvement was reached with samples containing 0.15–0.20 wt% CNTs/graphene oxide (at a 1:1 ratio), due to the interaction of the filler-layered structure with the polymer matrix [31]. Yue et al. [32] showed that CNTs coupled with graphene nanoplatelets (GnPs) at an 8:2 ratio led to a synergistic increase in the flexural properties of the epoxy matrix, which was caused by better dispersion of CNTs. Recently, it was confirmed that simultaneous addition of 0.17 wt% of amine-functionalized MWCNTs (NH_2_-MWCNTs) and 0.17 wt% of graphene nanoparticles resulted in the maximum improvements in fracture toughness (K_C_) and tensile strength at a testing temperature of −20 °C [33]. The synergistic effects resulted from the uniform dispersion of the nanofillers and the strong adhesion between nanofillers and epoxy. The modifier NH_2_-MWCNT was found to be very useful in conjunction with amine-functionalized graphene with respect to synergistic effects on the tensile strength and thermal stability of the epoxy matrix [34]. Tangthana-Umrung et al. [35] obtained a positive enhancement of the fracture toughness. The synergistic toughening of the hybrid epoxy base was attributed to the improvements in crack deviation and bifurcation, which resulted in a shorter crack path.

Carbon nanotubes were also combined with soft modifiers such as flexible polyurethane [36,37,38] or rubber particles [39,40,41] to improve the properties of the DGEBA matrix. Jia et al. [36] confirmed synergistic strengthening and toughening effects, specifically, the tensile, flexural and impact strengths, as well as fracture toughness (K_C_), in an epoxy-grafted polyurethane (EP-PU) modified with NH_2_-MWCNTs. The thermal stability and fracture toughness of epoxy resin were significantly improved through the simultaneous use of functionalized carbon nanotubes and carboxyl-terminated butadiene acrylonitrile [39]. Morphology analysis revealed the association of the plastic deformation zone with rougher surfaces.

Because polymeric materials are often used under different atmospheric conditions, Jen et al. [42] investigated the effects of temperature on the static tensile strength and fatigue resistance of epoxy/graphene/CNTs nanocomposites. The results showed that tensile strength and fatigue resistance decreased with increasing temperature.

Hybrid epoxy nanocomposites can be used, among other applications, in the aviation and automotive industries because of the outstanding performance properties of these materials. These lightweight materials can also be applied in packaging, construction and coatings. However, hybrid nanocomposites based on tougher polymers could be used in applications requiring impact and deformation resistance.

The purpose of the present study was to prepare hybrid epoxy composites utilizing either carbon nanotubes and polyurethane or montmorillonite, with the aims of improved mechanical and thermal properties. A synergistic toughening was expected as a result of the possible interactions between the modifiers and the epoxy matrix.

## 2. Experimental

### 2.1. Materials

The following ingredients were used in the present work:

Epoxy resin (Epidian 52 purchased from Sarzyna Co., Nowa Sarzyna, Poland), which has an epoxy number in the range of 0.510–0.550 mol/100 g and a viscosity between 400–800 mPa·s at 25 °C;Triethylene tetramine (trade name Z1, from Sarzyna Co., Nowa Sarzyna, Poland), which was used as a curing agent;Polyurethane prepolymer (Desmocap 12), produced by Bayer AG, Leverkusen, Germany;Carbon nanotubes, pyrolitically stripped platelets measuring D × L 100 nm × 20–200 µm, manufactured by Sigma Aldrich Co., St. Louis, MO, USA;Nanomer I.28E, nanoclay modified with 25–30 wt% trimethyl stearyl ammonium, produced by Nanocor Inc. Copenhagen, Denmark.

### 2.2. Preparation of Samples

#### 2.2.1. Epoxy-Based Composites with One Modifier

Epoxy resin was mechanically mixed with different amounts of polyurethane (PUR) for 10 min. CNTs and Nanomer I.28E nanoparticles were mixed with the epoxy matrix using a mechanical stirrer and an ultrasonic stirrer. For the Nanomer I.28E, the epoxy composition was mixed for 10 min with a mechanical stirrer, and this was followed by mixing using an ultrasonic stirrer for 75 min at maximum amplitude of 270 μm. The CNT was mixed for 10 min with a mechanical stirrer, followed by mixing with an ultrasonic stirrer for 8 h at a maximum amplitude of 270 μm. Then, 14 phr of curing agent was added to each mixture and mixing continued for 5 min. The compositions were poured into metal molds, and then cured for 24 h at room temperature and post-cured for 3 h at 80 °C.

#### 2.2.2. Hybrid Epoxy-Based Composites

The following hybrid epoxy composites were prepared: PUR/CNT and Nanomer/CNT. The epoxy resin was mixed with the modifiers using a mechanical stirrer, followed by ultrasonic mixing as previously described. The mixing time and sonication amplitude were defined from the maximum impact strength of the tested nanocomposites. In order to obtain the desired mechanical properties, the ingredients were incorporated into the matrix in the following order: PUR–CNT–curing agent or CNT–nanoclay–curing agent. Finally, 14 phr of curing agent was added, with an additional 5 min of mixing performed before the materials were poured into metal molds. The curing and post-curing were carried out as above. A schematic presentation of the composite preparation is shown below. The scheme of obtaining hybrid nanocomposites is shown in Figure 1 (below).

### 2.3. Evaluation of Mechanical and Thermal Properties

The determination of flexural properties was carried out in accordance with the relevant standards at a room temperature, and four samples were used for each data point. In addition, error bars have been added in figures.

Three-point Bending: The test was carried out on samples 10 cm long, 1 cm wide and 0.5 cm thick using a Zwick Roell machine, according to ISO-178-2019 [43]. The deformation rate was fixed at 5 mm/min.Charpy impact strength: The test was conducted with a Zwick Roell, on samples with the dimensions as described above and 1 mm of notch length, according to ISO-179-1:2023 [44]. The distance between the spans was 6 cm.Critical stress intensity factor (K_C_): Samples with dimensions and notch lengths identical to the samples used for strength were used for the test, which was carried out on a Zwick Roell device by means of ISO 13586:2018 [45]. The deformation rate was fixed at 5 mm/min. The parameter K_C_ was calculated as follows:
(1)$$K_{C}=\frac{3P\cdot L\cdot a^{1/2}}{2B\cdot w^{2}}\cdot Y$$
where *P* represents the load-at-break, *L* the distance between the spans, *a* the notch length, *w* the sample width, *B* the sample thickness and *Y* a geometrical factor which depends on the *a/w* ratio.Thermogravimetric Analysis: The test was performed using a Q500 thermogravimetric analyzer (TA Instruments, New Castle, DE, USA) in a nitrogen atmosphere, with a heating rate of 10 °C/min. and a temperature profile of 25–800 °C.Differential scanning calorimetry (DSC): The test was performed on a 1 Star System calorimeter (Mettler Toledo Warszawa, Poland) under a nitrogen atmosphere and with a scanning rate of 10 °C/min.

### 2.4. Evaluation of Structure and Morphology Analysis

Fourier transform infrared spectroscopy (FTIR) was used to show the functional groups present in the samples. The test was carried out on the Nicolet 6700 spectrometer (Thermo Fisher Scientific, Waltham, MA, USA), mode ATR with diamond crystal, 64 scans, resolution 4 cm^−1^.

The morphology of the samples was analyzed using a scanning electron microscope (SEM), the NovaNano SEM 450 microscope (The Netherlands, FEI company, Eindhoven, Th Netherlands).

## 3. Results and Discussion

### 3.1. Mechanical Properties

Figure 2 shows the effect of Nanomer nanoclay content on the impact strength (IS) of epoxy resin modified with 1 wt% carbon nanotubes (CNTs). It was observed that hybrid epoxy nanocomposite containing 0.5 wt% nanoclay and 1 wt% CNT showed a maximum increase in IS of approximately 70% and 20% above the IS of unmodified epoxy matrix and the nanocomposite with 1 wt% CNT. The IS of the ternary epoxy composite containing 0.5 wt% Nanomer and 1 wt% CNT (4.8 kJ/m^2^) was superior to those of the binary epoxy based on 1 wt% CNT (4.0 kJ/m^2^) and the sample based on 0.5 wt% Nanomer (2.7 kJ/m^2^), confirming the occurrence of a synergistic effect. The significant improvement in IS can be attributed to the exceptional mechanical properties of CNTs as well as their sufficient and homogeneous dispersion in the polymer matrix. The additional parameters to take into account, and factors which could have contributed to the improvement, would be the specific interfacial interactions between the modifiers with polymer matrix as well as the high degree of intercalation/exfoliation of the Nanomer in the matrix. Similar results were reported elsewhere with nanoclays [46,47]. Moreover, the impact strength decreased at higher nanoparticle contents, due most probably to the agglomeration of the nanoparticles, which constitute weak zones in the nanocomposites through which the resistance to crack propagation becomes rather weak and consequently, the IS shows a low value. Agglomeration of microparticles or nanoparticles is common at higher particle-loading levels in different systems [48,49].

The impact strength (IS) of epoxy resin modified with 1 wt% CNT is shown in Figure 3 as a function of the polyurethane (PUR) content. The maximum improvement in IS due to the synergistic effect was exhibited by the hybrid epoxy nanocomposite prepared with 7.5 wt% PUR and 1 wt% CNT. The improvement reached ~77.5% and ~85% compared to the nanocomposite containing 1 wt% CNT (without PUR) and epoxy with 7.5 wt% PUR (without CNT), respectively. Furthermore, it was demonstrated that the hybrid nanocomposite prepared with 5 wt% PUR and 1 wt% CNT had an IS of 6.7 kJ/m^2^, comprising a synergistic improvement in IS compared to both epoxy binary composites. The IS improvement may also result from the strong interactions between CNTs and the epoxy/PUR system. In addition, polyurethane may penetrate the porous structure of the CNT, which would explain the drastic enhancement in the IS of the epoxy nanocomposite. Similar results were presented in other studies with flexible polymeric chains and nanoclays [5,20,50,51]. It should be emphasized that the IS increased from 2.8 kJ/m^2^ for pure epoxy resin to a maximum of 5.1 kJ/m^2^ due to the addition of 2.5 wt% PUR, but only to 3.7 kJ/m^2^ with 7.5 wt% PUR. The improvement in IS can be attributed to the formation of an interpenetrating polymer network combined with a grafting reaction between the reactive groups of the matrix (-OH) and the polymer modifier (-NCO) [15,52].

The obtained results confirmed the occurrence of a synergistic effect in the IS of the ternary epoxy nanocomposite containing 0.5 wt% Nanomer and 1 wt% carbon nanotubes (CNT). The IS of the hybrid nanocomposite (4.8 kJ/m^2^) exceeded those of epoxy/CNT (4.0 kJ/m^2^) and epoxy/Nanomer (4.0 kJ/m^2^) by 20% and 75%, respectively, confirming the occurrence of a synergistic toughening of the epoxy matrix. The best impact resistance of the epoxy nanocomposite was obtained when the CNT and Nanomer were used simultaneously rather than when they were used separately. Positive toughening of brittle epoxy resin may result from specific interfacial interactions between nanofillers as well as transfer of the applied stress from the matrix to the nanofillers.

Figure 4 shows the effect of polyurethane (PUR) content on the stress intensity factor (K_C_) of an epoxy matrix containing 1 wt% CNT (a) and the K_C_ of epoxy resin containing 1 wt% CNT and different amounts of Nanomer nanoclay. It can be seen that the addition of 3.75–7.5 wt% PUR did not affect the values of the K_C_ factor, most probably due to the flexibilization of the epoxy achieved by the polymeric modifier. Furthermore, the incorporation of Nanomer nanoparticles did not lead to a hybrid nanocomposite with higher K_C_ values compared to the binary epoxy nanocomposite containing 1 wt% CNT. These results are contrary to those for impact strength (Figure 2 and Figure 3), although both IS and K_C_ evaluate the resistance of a material to crack propagation.

As the K_C_ did not demonstrate any improvement in the resistance to crack propagation in the tested samples, the fracture energy, which takes into account the contribution of stress and deformation, will be used instead.

The effect of Nanomer I.28E nanoclay content on the fracture energy of epoxy resin containing 1 wt% CNT is presented in Figure 5. The fracture energy was determined from the area under the load–deflection curve obtained during the crack propagation test and the evaluation of the critical stress intensity factor (K_C_). The energy required to fracture the hybrid epoxy prepared with 1 wt% CNT and 0.5 wt% Nanomer is more than ~15% higher than that required for the epoxy nanocomposite containing 1 wt% CNT and about ~75% compared to the epoxy matrix with 0.5 wt% Nanomer. In addition, the fracture energy value of the hybrid nanocomposite was higher than that of the pure matrix by more than 37.5%, thus demonstrating the toughening effects of the two incorporated modifiers. The positive effect of the simultaneous use of Nanomer nanoclay and carbon nanotubes is clearly confirmed, and the synergistic effect compared to binary nanocomposites may come from the interfacial interaction between the nanoparticles, but is also due to that between the latter and the matrix.

The fracture energy of epoxy resin containing 1 wt% CNT is shown in Figure 6 as a function of polyurethane (PUR) content. The energy increased from the 4 kJ/m^2^ of the unmodified epoxy matrix to 7.2 kJ/m^2^ for the hybrid epoxy nanocomposite based on 7.5 wt% PU and 1 wt% CNT, which represents an 80% improvement in energy. Nevertheless, a significant energy improvement of ~60% was obtained with 5 wt% PU/1 wt% CNT hybrid.

The fracture energy enhancement might be attributed to the presence of the flexible chain of the polymeric modifier (PUR), with which more free volume is created, leading to a more flexible hybrid nanocomposite sample, and a higher energy is required to break the sample. Similar results were reported with epoxy/polyurethane-based nanocomposites [36,53,54].

Table 1 presents the values of the critical stress intensity factor (K_C_) and the fracture energy of epoxy/CNT nanocomposites modified with different amounts of PUR (Table 1a) and Nanomer (Table 1b). It can be seen that the incorporation of polyurethane did not affect the fracture toughness parameter (K_C_), most probably due to the flexibilization of the epoxy matrix and the associated lower value of stress-at-break. However, the fracture energy, which is calculated from the contributions of the stress and the relative elongation, is a more appropriate metric to use to define the resistance to crack propagation in the samples. The brittle fracture energy is nothing other than the G_C_ for materials obeying the law of linear elastic fracture mechanics or the J_C_ for materials whose behavior is nonlinear [55]. A similar trend was demonstrated by the K_C_ of the epoxy/CNT/Nanomer ternary nanocomposites (Table 1b).

Table 2 presents the values of flexural strength and flexural energy-to-break of epoxy resin modified with 1 wt% CNT and different amounts of Nanomer (Table 2a) or polyurethane content (Table 2b). The addition of 0.5−2 wt% Nanomer 3.75−10 wt% did not induce any improvement in flexural strength compared to the epoxy/1 wt% CNT system. Similar results were reported in other studies [56,57,58]. This was attributed to the stress concentration in particle agglomerates and/or the formation of micro-voids, which act as weak points in the composites. Moreover, poor interactions or a lack of interactions between nanoparticles and a matrix may contribute to decrease in flexural strength. Interestingly, the trend in flexural strength is similar to that of the K_C_ parameter (Table 1) and can therefore be used to explain the decrease in the latter. Indeed, as expressed in Equation (1), the K_C_ which was evaluated under flexure is directly proportional to the applied flexural load.

The flexural strain-at-break of the epoxy resin filled with 1 wt% CNT is shown in Figure 7 as a function of the Nanomer nanoclay content. The strain-at-break increased from 2.6% for the unmodified matrix to 4.1% for the hybrid epoxy nanocomposite based on 0.5 wt% Nanomer nanoclay and 1 wt% CNT. The increase in strain-at-break reached about 60% and 25% compared to the pure epoxy matrix and that containing 1 wt% Nanomer, respectively. The improvement in the strain of the ternary epoxy nanocomposites might come from the strong physical interaction between the nanoparticles.

The effect of polyurethane (PUR) content on the flexural strain-at-break of the epoxy matrix modified with 1 wt% CNT is shown in Figure 8. As in the case of the fracture energy-at-break (Figure 6), the maximum increase in the flexural strain-at-break was obtained with a hybrid epoxy nanocomposite containing 7.5 wt% PUR. The strain-at-break improvement attained improvements of 90% and 50% compared to pure epoxy resin and matrix filled with 1 wt% CNT. The strain-at-break of the epoxy hybrid was higher (4.9%) than that of the binary epoxy/CNT (3.3%) but lower than that of the epoxy/PU blend (5.5%). The strain-at-break of the epoxy hybrid was higher than that of the binary epoxy/CNT blend but lower than that of the epoxy/PU blend. This result proved that the polyurethane was mainly responsible for the increase in the flexural strain-at-break, in addition to the impact strength (Figure 3) as well as the brittle fracture energy (Figure 5). It is well understood that, as discussed, the flexibilization or plasticization of the polymer samples for the above-mentioned properties leads to improvement of the performance properties.

Figure 9 and Figure 10 show the load–deflection curves of epoxy resin modified with CNT/PUR and CNT/Nanomer, respectively. In the case of the CNT/Nanomer systems, the curves are more linear, with a higher load-at-break. However, the CNT/PUR blends exhibited a non-linear behavior, along with a more pronounced ductility, which was caused by the flexible polymer chains. The energy-at-break levels of the epoxy nanocomposites were higher than those of the pure matrix and the blends based only on nanoparticles. From Figure 9, it can be seen that the blend based on 7.5 wt% PUR and 1 wt% CNT showed the highest deflection at break and energy-to-break, as compared to the other compositions and the pure epoxy matrix. This finding can explain the significant improvement of impact strength observed in the epoxy hybrid nanocomposites.

### 3.2. Evaluation of Thermal Properties

Table 3, below, shows the transition temperatures of epoxy composites modified with nanofillers, PUR or both modifiers. It can be noted that the addition of modifiers affected the glass transition temperatures of the prepared samples. The addition of nanofillers did not lead to drastic changes in the transition temperatures (only a few degrees). However, PUR significantly affected the Tp value, which increases with an increasing amount of polymeric modifier [59]. The addition of CNT, alone or combined with the other modifiers, had a significant effect on these transition temperatures (more than 30 degrees, compared to the pure matrix). Hybrid composites based on Nanomer and CNT showed similar glass transition temperatures compared to the binary sample modified only with CNT. For the sample containing 7.5 wt% polyurethane, a endothermic peak at 120 °C was observed. This may come from the post cross-linking reaction of the epoxy resin being the matrix of the composite. Based on the DSC analysis (Figure 11), no relationship was observed between the modifiers and the thermal characteristics of the produced materials. However, it is found that the additives used in the epoxy resin do not deteriorate the thermal properties of the obtained composites.

Figure 12 shows thermograms of epoxy–polyurethane mixtures in term of weight loss as a function of temperature. The heat resistance of the samples was defined at their 5% weight loss. The pure epoxy resin was characterized by two stages of decomposition, namely, at temperatures of 128 °C and 331 °C. Its thermal resistance was determined to be 192 °C, while its thermal decomposition took place in two stages. The first stage occurred at 200 °C and was associated with a 15% weight loss, while the second stage was noted at 358 °C, with a total weight loss of 93.5%. However, the addition of 1 wt% CNT delayed the occurrence of the first stage of decomposition (corresponding to a 5% weight loss) from 192 °C to 340 °C. The sample modified with 7.5 wt% polyurethane had properties similar to those of the 1 wt% CNT composite. The onset of the decomposition temperature of the EP/Nanomer blends was close to that of pure epoxy. Both hybrid blends modified with PUR/CNT have a starting point of the decomposition stage (5% weight loss) at a temperature of 205 °C. The relevant decomposition temperatures of the epoxy composites containing nanofillers and PUR are summarized in Table 4 below.

Based on the thermogravimetric tests carried out, the impacts of the additives used on heat resistance and the improvements in heat resistance as a result of these modifications are noticeable.

### 3.3. Structure and Morphology Analysis

FTIR spectroscopy was used to study the modification mechanism of epoxy resin matrix with polyurethane, CNT and Nanomer. The FTIR spectra of the epoxy matrix were compared with the FTIR spectra of selected epoxy composites containing modifiers significantly improving the mechanical properties. The changes in the FTIR spectra are shown in Figure 13.

As can be seen, there are peaks of similar intensity in all spectra. They come from vibrations of formations characteristic of epoxy resin, and the group frequencies have identical values. In all spectrograms (Figure 13), we can identify stretching vibrations of the C-H groups in CH_2_ and C-H of the aliphatic and aromatic systems occurring in the range of 2985–2870 cm^−1^. We also see a similar intensity of bands in the range of 1608 cm^−1^ (coming from C=C stretching vibrations) and also at the wave number 1509 cm^−1^ (C-C stretching) of the aromatic rings. At 1036 cm^−1^ we notice a signal coming from the C-O-C stretching vibrations of the ethers groups. Next, at the wave number of 770 cm^−1^ we have a small peak indicating a deformation in the vibrations of the CH_2_ groups. However, we notice some bands varying in intensity or disappearing in samples containing modifiers. We notice these differences at the peaks originating from the stretching vibrations of the O-H groups (3300 cm^−1^) and, to lesser extent, oxirane groups (900 cm^−1^ and 840 cm^−1^), which are characteristic of pure epoxy resin. The mentioned differences were observed in binary composites containing polyurethane or CNT as modifiers and in their hybrids. In samples containing 7.5 wt% PU, the intensity of the peaks associated with the vibrations of the O-H groups is half as much, as compared to pure epoxy resin [52,60]. We have a similar phenomenon when using 1 wt% CNT as a modifier, although it is not as clear. In the spectrograms of the mentioned compositions, we also observe a similar (to O-H) reduction in the intensity of the peaks corresponding to epoxy groups. This may suggest a reaction between the characteristic groups of the epoxy matrix and the reactive groups of polyurethane and the interaction with nanofillers. For hybrid nanocomposites we can observe that vibrations of the O-H groups is lower for the mixtures Nanomer/CNT and PUR/CNT. The vibrations of the O-H groups for the composite containing 0.5 wt% Nanomer and 1 wt% CNT is significantly lower, as compared to pure epoxy matrix and binary composite modified 0.5 wt% Nanomer. Both CNT/PUR hybrids shown in Figure 13 had lower vibration in the O-H group peaks. It was shown that the addition of 1 wt% CNT or 7.5 wt% of PUR caused the best interactions with the O-H groups, and thus was described as the optimum amount of each modifier to be used. Moreover, it can be observed that combining these two modifiers together or adding one of them to another modifier also causes significant improvement in the interactions with O-H groups. The confirmed interactions between the epoxy resin and the modifiers resulted in improved mechanical properties of the epoxy resin but also allowed the determination of the reinforcement mechanisms of the polymer matrix.

SEM images of the pure epoxy matrix and its composites are shown in Figure 14. As expected, the SEM image of the unmodified cured epoxy resin (Figure 14a) showed a smooth fracture surface with river-like patterns, typical of brittle thermosetting polymers. However, the nanocomposites based on carbon nanotubes (Figure 14b,c,e,f,h) show a wavy and rough surface with plastic deformations in addition to good dispersion of nanofillers, thus explaining the improvement of the mechanical properties. Similar observations were reported by other researchers [61]. The epoxy/PUR blends (Figure 14d,g) had a fracture surface with less intense but uniform roughness and less plastic deformation.

## 4. Conclusions

The following conclusions were drawn from the results obtained:This work confirmed the successful preparation of epoxy hybrid composites with improved mechanical properties: impact strength and brittle fracture energy were significantly increased compared to pure resin and binary systems. Indeed, the following synergistic effects have been obtained:
⁻The impact strengths of epoxy hybrid composites containing 1 wt% CNT and 5 wt% or 7.5 wt% of polyurethane and 0.5 wt% Nanomer and 1 wt% CNT increased by approximately 140%, 155%, and 70%, respectively;⁻The fracture energy for hybrid nanocomposites modified with PUR/CNT and Nanomer/CNT systems increased by approximately 80% and 50%, respectively;⁻The significantly improved mechanical and thermal properties of epoxy hybrids can be attributed to the uniform dispersion of modifiers in the matrix and the interfacial interactions between the ingredients. It is essential to create as many interactions as possible between the matrix and the modifiers, and within the latter, to eliminate agglomerates in order to generate the optimal mechanical properties of the hybrids produced.The addition of polyurethane and nanomodifiers increased the thermal stability of epoxy composites. The addition of flexible polyurethane chains increased the glass transition temperature as well as the softening point and the temperature range of use of the epoxy nanocomposites containing nanofillers.Interactions between the epoxy matrix and the added modifiers through the O-H groups of the epoxy resin with PUR and CNT were confirmed.Hybrid epoxy composites with improved performance properties can be safely used in harsh environments or used as advanced composite materials in the aerospace and construction industries.

## Figures and Tables

**Figure 1 polymers-17-00158-f001:**
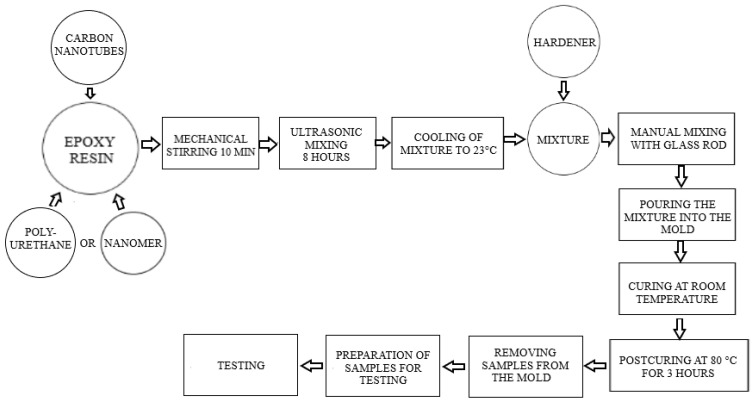
Scheme for obtaining hybrid composites.

**Figure 2 polymers-17-00158-f002:**
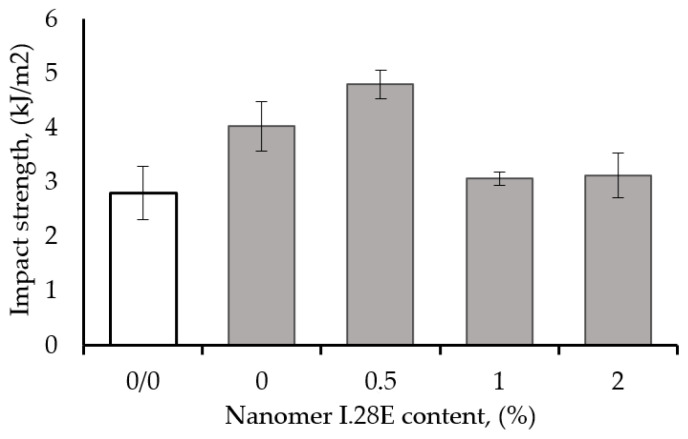
Effect of Nanomer content on the impact strength of epoxy resin modified with 1 wt% CNT.

**Figure 3 polymers-17-00158-f003:**
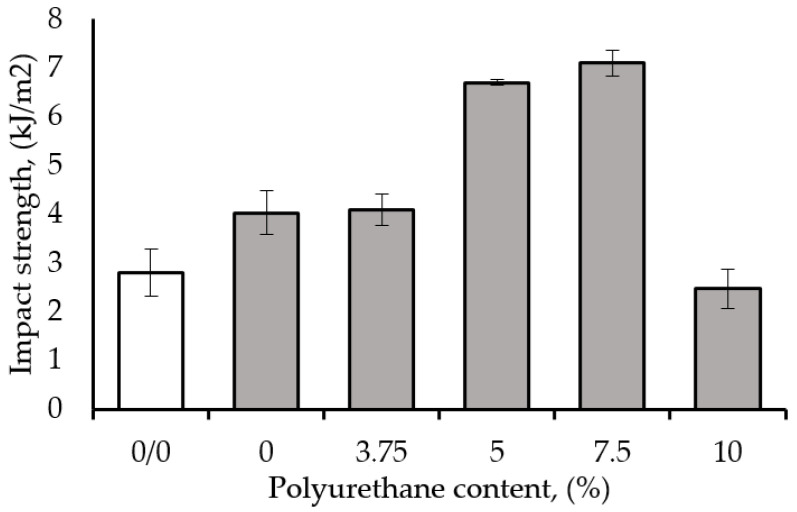
Effect of polyurethane content on the impact strength (IS) of epoxy resin modified with 1 wt% CNT.

**Figure 4 polymers-17-00158-f004:**
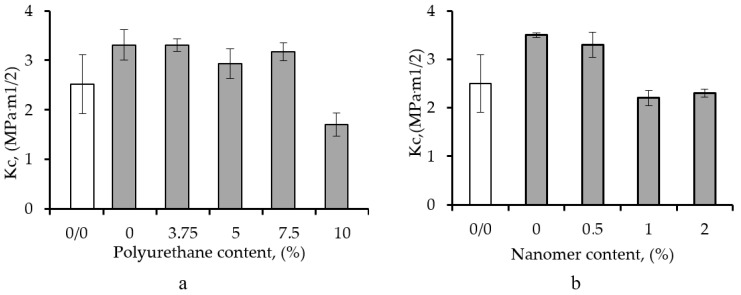
Stress intensity factor (K_C_) of epoxy matrix modified with 1 wt% CNT, as a function of polyurethane content (**a**) and Nanomer content (**b**).

**Figure 5 polymers-17-00158-f005:**
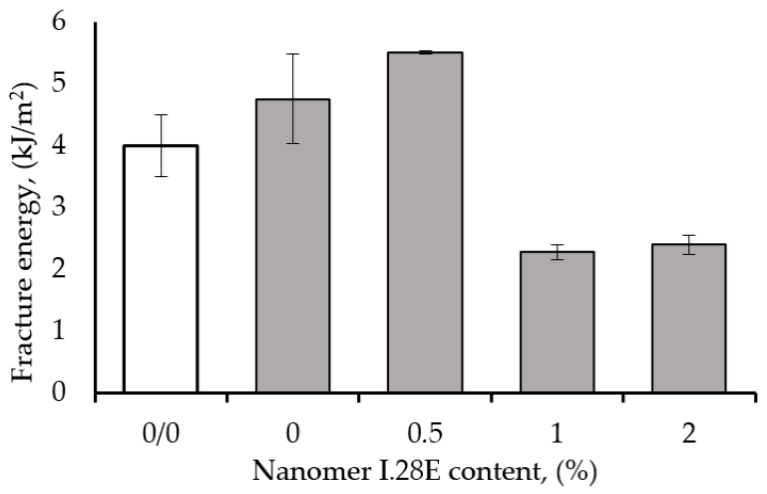
Fracture energy of epoxy resin modified with 1 wt% CNT as function of Nanomer nanoclay content.

**Figure 6 polymers-17-00158-f006:**
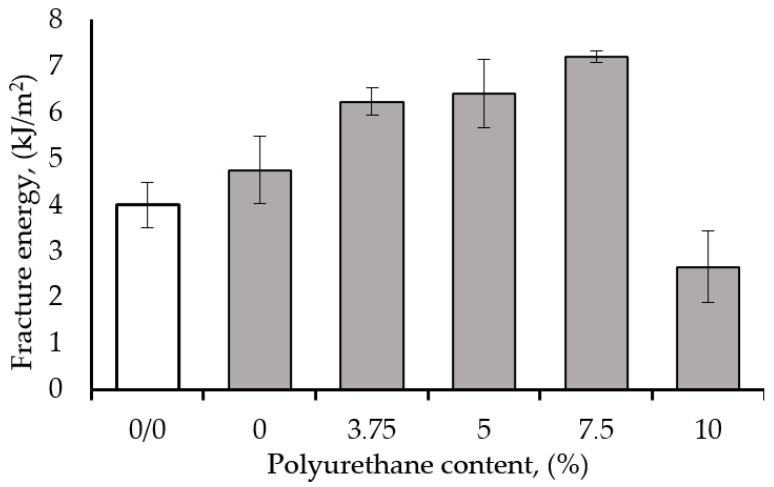
Fracture energy of epoxy resin modified with 1 wt% CNT as function of polyurethane content.

**Figure 7 polymers-17-00158-f007:**
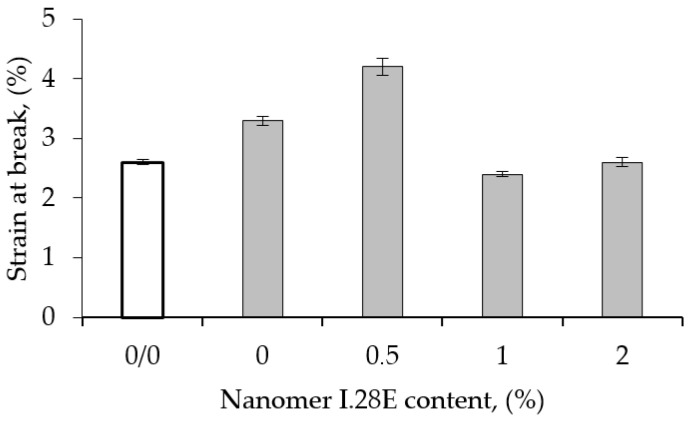
Flexural strain-at-break of epoxy matrix modified with 1 wt% CNT as function of Nanomer content.

**Figure 8 polymers-17-00158-f008:**
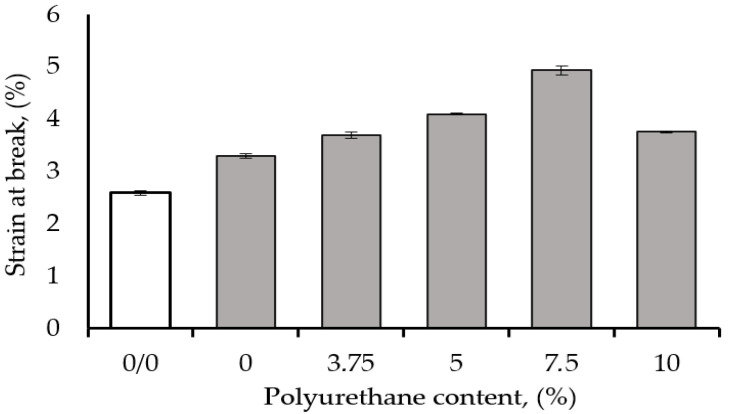
Effect of polyurethane content on the flexural strain-at-break of an epoxy matrix modified with 1 wt% CNT.

**Figure 9 polymers-17-00158-f009:**
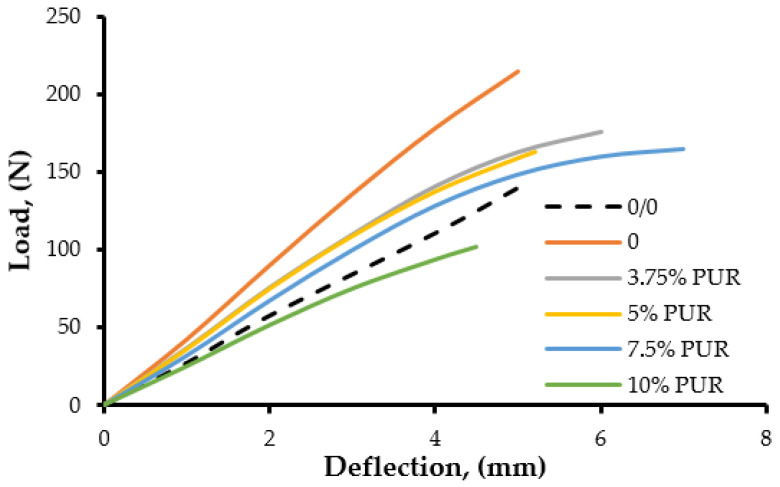
Load–deflection curves of epoxy resin modified with 1% CNT and different amounts of polyurethane.

**Figure 10 polymers-17-00158-f010:**
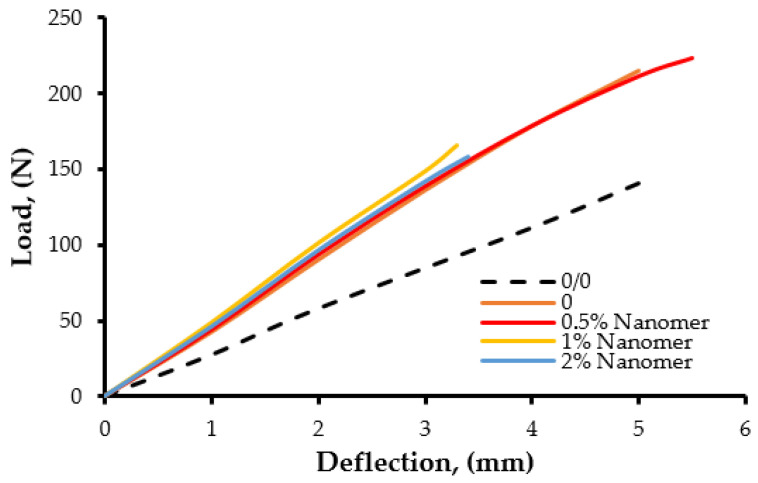
Load–deflection curves of epoxy resin modified with 1% CNT and different amounts of Nanomer I.28E.

**Figure 11 polymers-17-00158-f011:**
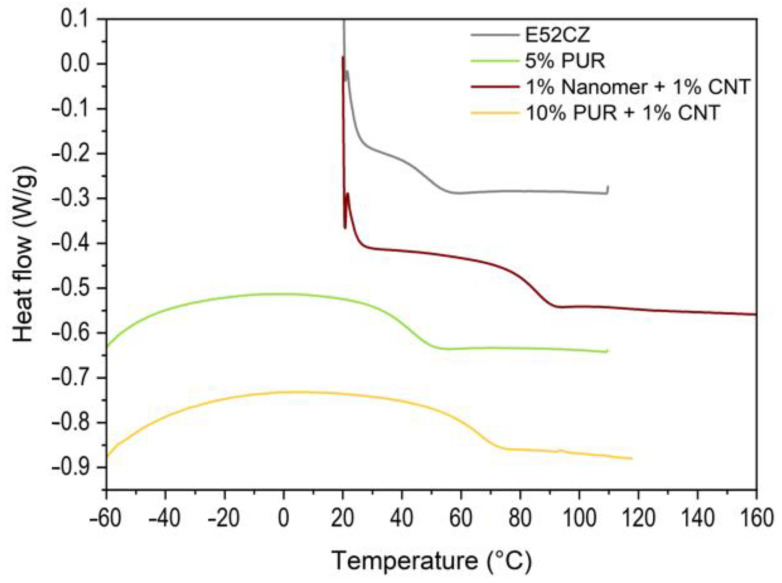
DSC thermograms of epoxy resin (second heating), modified with different modifier systems.

**Figure 12 polymers-17-00158-f012:**
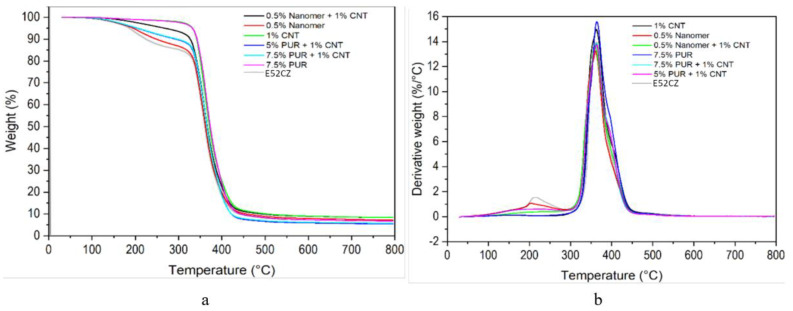
TGA (**a**) and dTG (**b**) thermograms.

**Figure 13 polymers-17-00158-f013:**
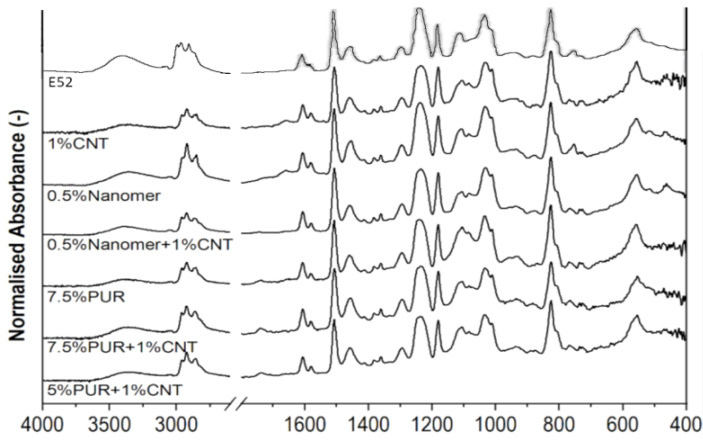
FTIR spectra of selected epoxy compositions.

**Figure 14 polymers-17-00158-f014:**
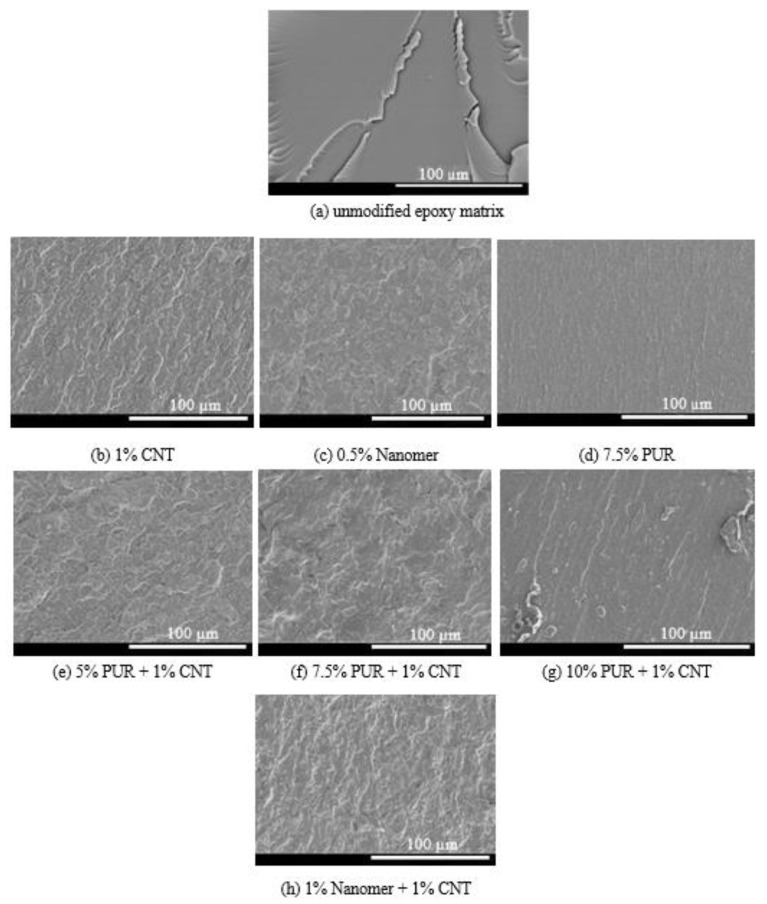
SEM micrographs of pure epoxy matrix and epoxy composites.

**Table 1 polymers-17-00158-t001:** Values of critical stress intensity factor (K_C_) and brittle fracture energy of epoxy/CNT nanocomposites based on polyurethane content and (a) Nanomer nanoclay content (b).

PUR Content (wt%)	K_C_ MPa.m^0.5^	Brittle Fracture Energy (kJ/m^2^)	Nanomer Content (wt%)	K_C_ MPa.m^0.5^	Brittle Fracture Energy (kJ/m^2^)
0/0	2.8	4.0	0/0	2.5	4.0
0	3.2	4.8	0	3.5	4.8
3.75%	3.3	6.2	0.5	3.3	5.5
5%	2.9	6.4	1	2.2	2.3
7.50%	3.2	7.2	2	2.3	2.4
10%	1.8	2.7			
a			b		

**Table 2 polymers-17-00158-t002:** Flexural strength and flexural and flexural energy-to-break of hybrid epoxy resin containing 1 wt% CNT and different amounts of Nanomer (a) or polyurethane (b).

Nanomer Content (wt%)	Flexural Strength (MPa)	Flexural Energy to Break (kJ/m^2^)	Polyurethane Content (wt%)	Flexural Strength (MPa)	Flexural Energy to Break (kJ/m^2^)
0/0	73.0	17.5	0/0	73.0	17.5
0	106.0	14.5	0	70.0	14.5
0,5	94.9	15. 8	3.75	71.0	14.1
1	74.7	6.3	5	67.9	10.6
2	68.4	4.4	7.5	71.7	18.6
			10	42.0	5.9
a			b		

**Table 3 polymers-17-00158-t003:** DSC thermograms of selected epoxy compositions.

Sample	m_0_ [mg]	T_p_ [°C]	T_g1_ [°C]	T_p1_ [°C]	T_g2_ [°C]
Pure epoxy matrix	10.17	56	44	----	43
1% CNT	9.65	---	82	92	82
0.5% Nanomer	10.24	58	47	---	50
0.5% Nanomer + 1% CNT	10.45	57	79	---	77
1% Nanomer + 1% CNT	10.11	91	83	---	85
5% PUR	9.86	48	45	55	46
7.5% PUR	10.39	121	68	82	78
10% PUR + 1% CNT	10.19	56	53	78	63
7.5% PUR + 1%CNT	10.58	---	50	---	54
5% PUR + 1%CNT	9.98	67	48	52	54

m_0_ [mg]—Initial weight of the sample; T_p_ [°C]—Temperature of endothermic peak, in some curves very small; T_g1_ [°C]—T_g_ midpoint extracted from first heating. T_p1_ [°C]—Temperature of endothermic peak (relaxation, after T_g1_). T_g2_ [°C]—T_g_ midpoint extracted from second heating.

**Table 4 polymers-17-00158-t004:** Temperatures of the degradation of epoxy nanocomposites modified with different types of modifier systems.

Sample	Thermal Stability T_ST_ [°C]	Peak 1		Peak 2		m_T_ [%]
		T_1_ [°C]	Δm_1_ [%]	T_2_ [°C]	Δm_2_ [%]	
Pure Epidian 52	192	199	6.73	357	90.74	93.52
1% CNT	340	163	1.29	363	90.27	91.55
0.5% Nanomer	194	202	12.41	360	80.42	92.83
0.5% Nanomer + 1% CNT	272	222	5.75	359	85.86	91.61
7.5% PUR	340	151	1.23	364	91.95	93.18
7.5% PUR + 1% CNT	208	231	9.29	363	84.88	94.17
5% PUR + 1% CNT	201	226	9.35	363	85.15	94.50

Δm_1_ [%]—The weight loss at T_1_ temperature at dTG peak 1; Δm_2_ [%]—The weight loss at T_2_ temperature at dTG peak 2; m_T_ [%]—The total weight loss.

## Data Availability

The original contributions presented in this study are included in the article. Further inquiries can be directed to the corresponding author.

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
