# Peer review of "Ternary Epoxy Nanocomposites with Synergistic Effects: Preparation, Properties Evaluation and Structure Analysis"

_polymers, 2025, doi:10.3390/polym17020158_

Round 1

Reviewer 1 Report

Comments and Suggestions for Authors

This paper dealt with the preparation and evaluation of the properties of hybrid epoxy composites based on polyurethane (PUR) and solid nanoparticles (Nanomer nanoclay and carbon  nanotubes).

The authors obtained a number of new results.

There are no fundamental comments, but I would like the authors to write in the manuscript their assumptions as to why it is with a certain content of fillers, PU that a sharp improvement in the properties of the composite is observed.

Changes in glass transition temperature are not usually specified in %. It is better to indicate them as a difference.

To remove questions about the reasons for the appearance of additional peaks on the DSC thermograms, the authors should have reheated the samples. The obtained value of the glass transition temperature of the hybrid composite during reheating would have been more convincing in drawing a conclusion about the effect of the introduced additives on this characteristic.

Author Response

Thank you for your detailed and insightful review of my article. I truly appreciate the time and effort you dedicated to providing such comprehensive comments, which have significantly enhanced the final version. Please find attached a document with our responses to the reviewer's comments and the corresponding revisions made to the article. I hope our answers will be satisfactory.

Kind regards,

Patryk Suroń

Reviewer 2 Report

Comments and Suggestions for Authors

A manuscript entitled “Ternary Epoxy Nanocomposites with Synergistic Effects: Preparation, Properties Evaluation and Structure Analysis” is well written and structured by the authors. The manuscript may be accepted after incorporating minor modifications in the manuscript. The changes required in the manuscript is as follow:

·       Authors must incorporate the motivation/need in 2-3 sentences in abstract before explaining current work.

·       Abstract is lengthy keep it short, precise and catchy.

·       The starting of Introduction section with “Due to the” looks weird. Do the needful to correct it.

·       Introduction section lags in connectivity, author needs to enhance the connectivity of introduction section by rearranging the information provided.  

·       Authors also needs to highlight the novelty of manuscript, in current form it’s hard to identify.

·       In introduction section different fonts style and size used along with underline which make it worst. Do the needful to correct it.

·       The level of English needs to be uplift, currently used style is very poor.

·       Application part is completely missing in introduction section. Authors must incorporate it.

·       Author must add proper details of composite processing along with schematic or real time images.

·       No details were given for ASTM standards and how many samples tested in which conditions?

·       Author must incorporate stress strain curves for flexural testing

·       SEM scales not visible in Figure 12 redraw the scales. If author wants to enrich SEM section use the following publications as reference: Tarun Nanda et al 2019 Mater. Res. Express 6 065061 and  https://doi.org/10.1080/03602559.2010.531427

·       Rewrite the conclusions precisely by removing repetition as discussed in abstract and discussion part. Only give the crux/ key outcomes having significance for scientific community along with future road map.

Author Response

(The authors gave the same response as above.)
